# Alignment of Air Pollution Exposure Inequality Metrics with Environmental Justice and Equity Goals in the United States

**DOI:** 10.3390/ijerph21121706

**Published:** 2024-12-21

**Authors:** Sarah Chambliss, Natasha Quynh Nhu Bui La Frinere-Sandoval, Corwin Zigler, Elizabeth J. Mueller, Roger D. Peng, Emily M. Hall, Elizabeth C. Matsui, Catherine Cubbin

**Affiliations:** 1Department of Population Health, The University of Texas at Austin Dell Medical School, Austin, TX 78712, USA; 2Steve Hicks School of Social Work, The University of Texas at Austin, Austin, TX 78712, USA; 3Department of Biostatistics, Brown University School of Public Health, Providence, RI 02903, USA; 4School of Architecture, The University of Texas at Austin, Austin, TX 78712, USA; ejmueller@austin.utexas.edu; 5Department of Statistics and Data Sciences, The University of Texas at Austin, Austin, TX 78712, USA; 6Center for Health and Environment: Education and Research, The University of Texas at Austin Dell Medical School, Austin, TX 78712, USA; 7Department of Pediatrics, The University of Texas at Austin Dell Medical School, Austin, TX 78712, USA

**Keywords:** air pollution, environmental justice, environmental equity

## Abstract

A growing literature within the field of air pollution exposure assessment addresses the issue of environmental justice. Leveraging the increasing availability of exposure datasets with broad spatial coverage and high spatial resolution, a number of works have assessed inequalities in exposure across racial/ethnic and other socioeconomic groupings. However, environmental justice research presents the additional need to evaluate exposure inequity—inequality that is systematic, unfair, and avoidable—which may be framed in several ways. We discuss these framings and describe inequality and inequity conclusions provided from several contrasting approaches drawn from recent work. We recommend that future work addressing environmental justice interventions include complementary “Exposure-driven” and “Socially weighted” metrics, taking an intersectional view of areas and social groups that are both disproportionately impacted by pollution and are impacted by additional health risks resulting from structural racism and consider implications for environmental justice beyond distributional equity.

## 1. Introduction

Although overall air pollution levels have declined in the United States (US) over the past four decades [1], the distribution of exposure across the US population remains unequal, with systematic differences in exposure remaining among racial/ethnic groups and by socioeconomic status (SES) [2,3,4,5]. This disparity in exposure among social groupings has been raised as an issue of environmental justice (EJ), defined by the Environmental Protection Agency (EPA) as “the fair treatment and meaningful involvement of all people regardless of race, color, national origin, or income, with respect to the development, implementation, and enforcement of environmental laws, regulations, and policies”. The EJ movement grew out of decades of community activism and the civil rights movement and was in reaction to environmental racism or the intentional policies and decisions that historically targeted poor communities of color for the siting of undesirable and harmful land uses, such as polluting industries [6,7,8]. In addition to being a human rights issue, environmental justice is a public health issue, given the many different harmful health impacts of environmental exposures to the population overall and to health disparities between social subgroups. Governmental bodies have recognized this by issuing multiple Executive Orders (e.g., EO 12898) and statutes directing federal agencies in various ways to address the harms of environmental injustice [9], and environmental laws exist at local levels as well (e.g., state, municipal). While important, the impact of these efforts is not well understood, partially because of differences in the methods for measuring exposure inequities.

Environmental justice has been described as having three elements: distributive justice, referring to the distribution of health hazards across the population; recognitional justice, referring to acknowledgement and respect for multiple perspectives and ways of knowing; and representational justice, ensuring equitable participation in decision-making, including the concept of procedural justice, relating to the institutional processes of the state [10]. Documenting air pollution exposure in the US as an area of environmental injustice requires more than a calculation of inequality, i.e., a quantitative estimate of differences in exposure across the population [11,12]. It additionally requires an evaluation of whether those inequalities are unjust; such inequality is described as inequity in the context of public health, with the additional qualities of being systematic, unfair, and avoidable [13,14,15]. Much of the literature examining air pollution exposure in the context of EJ focuses on the quantitative elements of distributive justice and furthermore may not explicitly identify whether documented inequalities are inequitable. Recognitional and representational justice are often even less explicitly addressed, although many works refer implicitly to concepts of procedural justice in describing conclusions and implications of their findings. In addressing issues of EJ, a quantification of exposure requires more than population-level summary statistics, as the issue of EJ presents a complex objective: decreasing exposure in a fair and equitable way. The requirement of the “fair treatment… of all people” presents the need for both distributive and procedural equity; the pollution exposure and resulting health burden should not be disproportionately distributed over a subset of the population, and those populations most affected by pollution exposure should be included in the decision-making process for exposure remediation efforts [16,17]. We seek to draw attention to these concepts of inequity and justice in a representative selection of the recent air pollution exposure literature and promote a more explicit consideration of EJ concepts in future work.

There is a growing literature outlining related practices for integrating the principles of EJ into research programs. A review by Van Horne et al. (2023) proposes a framework for the application of EJ principles in the conduct of exposure science, including the collection and handling of exposure data [18]. Gardner-Frolick et al. (2022) provide guidelines for selecting air pollution exposure data appropriate to the EJ context [19]. Specific to the practice of risk assessment and management, Levy (2021) recommends criteria and processes for selecting measures of exposure and health inequalities in environmental legislation decision-making, re-emphasizing several concepts established in older reviews of inequality metrics [11,16]. We aim to build further on the subject of metric selection and interpretation by identifying usage patterns in the recent air pollution literature and examining the orientation of these studies relative to the previously mentioned concepts of equity and justice. Many metrics have been used to quantify differences in air pollution exposure across subpopulations in the US, frequently with the intention of examining exposure inequity for racial/ethnic and income groups. The selection of an appropriate metric is not trivial; both the information represented by a chosen metric and the interpretation of and conclusions drawn from that metric typically only present a partial view of distributive and procedural EJ goals.

Given the increased scrutiny of EJ in air pollution exposure and an imperative to move on to actions to address it, paired with the new availability of community air quality (AQ) monitoring and large scale AQ prediction data, it is important to examine the pros and cons of different common metrics of inequities in air pollution exposure [20]. Therefore, the objectives of this commentary are threefold: to examine and discuss several conceptual framings represented in the recent air pollution EJ literature and how they align with the choice of inequality/inequity metrics, summarized in Table 1; to illustrate the range of inequality/inequity metrics and their conceptual alignments within a selection of recently published manuscripts; and to provide recommendations for selecting and combining metrics to match the policy goals of different types of interventions.

## 2. Conceptual Framings for Air Pollution Exposure Inequality Metrics

Below, we discuss four conceptual framings for exposure inequity. All four framings assume an evaluation of progress towards policy goals, but the specific goals sit along a spectrum of exposure equity ideals, ranging from to lowering pollution for everyone to reducing both absolute and relative inequalities in exposure to identifying optimal target sources or areas to maximize improvements in equality to lowering exposure for specific groups or communities that are the most disproportionately and inequitably burdened. For each framing, we describe the underlying policy goal, typical metrics aligned with evaluating progress towards that ideal, and the potential limitations of that framing in presenting a complete view of EJ, illustrated in Figure 1. We also consider extensions of each of these framings for health disparities.

### 2.1. Equal Improvement

The first framing (“Equal Improvement”) treats lower overall exposure as the primary goal. This is the least centered on EJ among the four framings because it neither seeks to address historically high exposure for specific subgroups—racial, socioeconomic, or geographic groups—nor does it integrate any social context informing the cause of higher exposure for particular groups. Instead, this framing presents equality as an equivalent reduction in exposure for minoritized populations and non-minoritized populations or for lower- vs. higher-income (or -SES) groups, with “low income” defined by the ratio of household earnings to the federal poverty level. Given the documented persistence of exposure inequality by race/ethnicity and income, equivalent reductions leave those inequities in place and thus do not advance distributive EJ goals, nor do they contribute to representational or procedural goals. Typical analytic approaches compare summary statistics for the total population across different years to quantify whether the goal of exposure reduction has occurred. If statistics are stratified by racial/ethnic or income/SES group (e.g., mean or median air pollution levels), the goal of “equal improvement” is achieved if the absolute change in those summary statistics over time is equivalent across subgroups. This typical approach oversimplifies exposure inequity by collapsing the experience of a group into a single summary metric that does not reflect heterogeneity within groups. This type of analysis, however, usually provides a baseline that research studies build on with the concepts discussed below.

### 2.2. Equality

The second framing (“Equality”) aims for equivalent levels of exposure across the population, which requires greater exposure reductions for those starting at a higher baseline exposure. When this goal is blind to subgroups within the population, it is a weaker expression of distributional equity—that it is fair that every person experiences the same level of air pollution exposure—but increases its justice alignment as subgroups defined by minoritized race/ethnicity or low income are identified as having systematically higher exposures. Within this equity framing, the objective is an improvement in air quality to an equally low level of exposure for all racial/ethnic or income groups with a corresponding policy goal of group-to-group equality, without preference for a particular group. Although increasing exposure for the least exposed group could theoretically increase equality, this is of course not relevant in practice nor desirable.

The most purely equality-focused type of metric is a single-value descriptor of the deviation of the distribution of exposure values across the population from a state of perfect equality (identical exposure across individuals). Different formulae may be used to express this deviation, with variations provided, such as the Atkinson index, Gini index, mean log deviation, or Theil entropy, with the specific mathematical attributes of these various options described in depth in previous reviews [11,16,27]. The basic application of this type of inequality metric considers the distribution of exposure across the entire population, which ignores the social context determining equity. However, several of these metrics allow for decomposition into between- vs. within-group inequality measures, and this introduces a more strongly equity-aligned comparison among defined groups.

Single-value inequality metrics may be complemented by pairwise comparisons of population-weighted summary statistics (e.g., mean, median) across different racial/ethnic groupings or income strata. Among works investing inequality, there is a range in the extension of these comparisons to equity. Some studies intentionally avoid integrating social context (e.g., structural and institutional discrimination) to predefine the key comparison pairs, opting to emphasize the gap between the highest- and least-exposed groups, while some works make decisions in the presentation of results that implicitly reflect social context, for example choosing to focus the primary results on the pairwise comparison of non-Hispanic white and Black racial/ethnic groups (and this extends into the social weighting framework discussed below). Because the generalization of the experience of a group into a single summary metric (e.g., mean exposure level) ignores variation in exposure within the group, quantifications of inequality may include a number of summary metrics, such as the 75th or 90th percentile. However, with the introduction of more metrics, questions arise such as the following: Is there a difference in the importance of the magnitude in the difference in group means or group extremes (e.g., 95th percentiles)? To further summarize inequality, is it appropriate to count how many times a subgroup is in the highest exposure level? A remaining limitation of these summary metrics in addressing equity is the treatment of all members of a given racial/ethnic group as interchangeable—ignoring geographic, cultural, and socioeconomic differences among members of the same racial/ethnic group—and any change over time only describes the experience of the group as a whole and not of individuals within the group.

As extended to public health, absolute inequality in exposure provides a crude indicator for the degree of disparity in adverse health outcomes. Exposure–response functions can be applied to estimate disparity in specific diseases attributable to air pollution exposure. Similarly, the relative difference in exposure may provide a conservative estimation of the relative differences in health outcomes. However, differences in baseline rates and differences in vulnerability related to race and ethnicity (by which we mean, here and below, the impacts of structural racism that shape social determinants of health) typically widen differences between groups, as is discussed further below.

### 2.3. Exposure-Driven Prioritization

The third framing (“Exposure-driven”) is one in which the priority is the group that is most exposed, referred to in some works as the “maximin” [11]. While a pure maximin strategy could ignore EJ concerns, prioritizing the highest-exposed communities regardless of their relative social status, a more EJ-centric application of this framing considers the representativeness of the racial/ethnic or socioeconomic composition of the highest-exposed group relative to the total population. This approach extends the “Equality” framing to address the equity ideal that there should not be a systematic racial/ethnic or SES difference in who is most exposed.

Metrics aligned with exposure-driven prioritization follow two general methods of evaluating the “most exposed” using either population or geographic areas as the units of analysis. With a population focus, an exposure level is assigned to each individual to define a high-exposure subgroup (e.g., the top 10%, the top 50%, etc., in the exposure distribution) and the degree of inequity is expressed as the difference in the representation of a particular racial/ethnic group or other population group (e.g., defined by income) in the high-exposure category vs. that in other exposure categories. For areal units (e.g., census tracts, neighborhoods, regions), the high-exposure areas are those where the concentrations are highest. Inequity could be evaluated as the difference in the demographic composition of the highest-exposure areas vs. elsewhere. Because high-exposure areas are fixed in time, while the specific members of the high-exposure group are not, areal units also allow inequity to be evaluated as a slower improvement in high-exposure areas relative to other areas. If additional information about health vulnerability is built into the impacts calculation, the benefit considered could include health disparities in addition to exposure disparities, which is more nuanced than distinguishing between simply “high-minority” areas and “high-vulnerability” areas.

### 2.4. Socially Weighted Prioritization

The final framing approach (“Socially weighted”) takes into account multiple characteristics impacted by structural racism that shape social determinants of health (e.g., considering low-income members of a racialized group), akin to an intersectional approach, when determining where changes in exposure improve equity the most. Building on the “Equality” concept that greater improvements are needed for higher-exposure groups, it further centers on EJ by using context beyond air pollution exposure to identify a population that is more vulnerable to the health effects of pollution exposure. A basic goal in this approach is that the vulnerable population should not experience systematically higher exposure than the rest of the population. Rather, vulnerable populations should experience lower exposure because the health impact on them is greater due to the externally caused vulnerability. Other individual characteristics affecting susceptibility to pollution-related health impacts, including sex, age, pregnancy, etc., may be considered when identifying vulnerable groups but do not reflect the social prioritization of historically marginalized populations. Identifying socially vulnerable populations can improve distributional justice and also facilitate progress in procedural justice by directing policy makers and regulators to the places and people who require increased representation in the policy-making process.

As above, typical metrics depend on how the vulnerable population is defined. Using persons as the unit of interest, groups may be defined based on a composite of characteristics often taken from census data. At its most basic level, this is a combination of demographic and SES characteristics, such as stratifying by both race/ethnicity and income. More sophisticated social vulnerability metrics use a composite of financial security, access to healthcare, other environmental exposures, etc., which may or may not explicitly include race/ethnicity. Like above, typical exposure metrics are summary statistics of exposure across the populations of high vulnerability vs. low vulnerability, including mean/median levels and sometimes exposure percentiles. While this approach addresses a limitation noted above by integrating the outside context into high-level population groupings based on census data, the reliance on a few summary metrics still provides a narrow view of how members of a vulnerable group experience pollution exposure. An important contrasting approach is using areas as the unit of interest, designating specific census units (block, block group, tract) or neighborhoods defined in other ways as vulnerable or of EJ’s concern based on demographic and environmental factors. Inequity is quantified by comparing air pollution exposure levels among discrete groupings of “more vulnerable” vs. “less vulnerable” census/neighborhood areas. Alternatively, inequity could also be quantified as a correlation between one or more continuous demographic/SES metrics and the pollution exposure observed in a census/neighborhood area. A particular strength of this approach is that it lends itself to tracking the same places over time, identifying inequity as high in vulnerability or with more predominantly minoritized areas ranking higher in relative pollution levels over time.

## 3. Diverse Approaches to Quantifying Exposure Inequality in Prior Publications

The choice of metrics used to quantify pollution exposure inequalities in air pollution exposure analyses expresses a set of EJ ideals which may not be explicitly stated in the work. Below, we discuss a range of complementary methodologies applied across a selection of US-based air pollution exposure studies that document air pollution inequalities/inequities. These works were chosen to illustrate the range of metrics and approaches applied in current EJ analyses. This is not a comprehensive review, and thus we do not include all studies addressing inequality and environmental injustice in air pollution exposure, nor do we address all aspects of these works, but we instead focus on elements of particular relevance to the framings described above.

We categorized the selection of papers chosen for this review by the type of study design: cross-sectional (*n* = 4), retrospective longitudinal (*n* = 3), and policy impact projection (*n* = 3). All cross-sectional and retrospective longitudinal studies are based on observational data—estimates of exposure from measurements of outdoor concentrations—while the policy impact projections rely on simulations of future conditions using physicochemical modeling. The cross-sectional and policy projection analyses include both urban-scale and national-scale evaluations, while the retrospective longitudinal analyses evaluate conditions over the entire continental US.

These studies cover a variety of air pollution species, with the most common being fine particulate matter (PM_2.5_) and nitrogen dioxide (NO_2_). Some studies include other pollutants regulated under the US EPA Clean Air Act (sulfur dioxide, SO_2_; carbon monoxide, CO; PM_10_, a broader size category of particulate matter; and ozone, O_3_) and, rarely, other measures of particulate matter (ultrafine particle count, or UFP). The methods used to calculate common metrics are described in Figure 2, including those used to summarize population-scale exposure (e.g., population-weighted mean, median, quantiles) as well as inequality indices used to estimate the degree of deviation from population-wide exposure equality (e.g., the Atkinson index, the Gini index). The studies also vary in the language and specific groupings used for race and ethnicity and “low income”, so some inconsistencies remain in the terminology used to describe race/ethnicity and income strata.

### 3.1. Cross-Sectional

Cross-sectional observational studies describe conditions for one point in time (e.g., one year). While this design is the most limited in its ability to support conclusions about the causality or the origin of conditions of inequality, the selections below make use of a combination of metrics to inform issues of both distributional and procedural equity.

The national-scale cross-sectional study we include, authored by Clark et al. (2014), documented exposure to NO_2_ in the year 2006, focusing on the questions of which race/ethnicities were most exposed and whether the differences by race/ethnicity were smaller than differences by income or other SES-based groupings [2]. The primary exposure metric was population-weighted mean concentrations by race/ethnicity, poverty status, income, education, and age. In secondary analysis, they used linear regression to model NO_2_ as a function of income stratified by race/ethnicity and included an evaluation of population-wide inequality using the Atkinson index. The authors reported statistically significantly higher exposure for “disadvantaged and historically disenfranchised groups”, with a lower population-weighted exposure for the non-Hispanic white population than for any other race/ethnicity, even when accounting for income. Further focusing on racial/ethnic disparities in the context of income, they reported that NO_2_ disparities by race (controlling for income) were twice the magnitude of income-based disparities (controlling for race). Low-income, nonwhite children and elderly people were the most disproportionately exposed. The Atkinson index and group-weighted summary statistics provided “Equality”-aligned context, but the analysis overall was positioned as “Socially weighted”; the cross-stratification and the probing of the competing or modifying effects of race/ethnicity with income interrogated which subgroups within may be most burdened by air pollution and thus should be given priority in policy.

Stuart et al. (2009) investigated exposure inequalities across racial/ethnic groups in Tampa, Florida, and its surrounding county, quantifying within-county differences in proximity to types of pollution sources as well as to air quality monitoring sites [21]. The authors applied a novel subgroup inequality metric: the log(ratio) of the fraction of the population of a racial/ethnic group living within a specified distance of defined types of pollution sources. A positive value indicated a higher level of inequity (members of this group disproportionately resided inside the high-exposure zones) and a negative value indicated that members of this group disproportionately resided outside the high-exposure zones. Based on these subgroup inequality metrics, Black, Hispanic, and high-poverty groups were shown to disproportionately live closer to hazards and farther from monitoring sites. Although the subgroup inequality metric investigated high exposures in areas around sources, it was less aligned with the “Exposure-driven” framework and more aligned with “Equality” because the analysis focused on comparisons among different racial/ethnic groups rather than drawing conclusions about which areas or sources to target for mitigation.

Chambliss et al. (2021) investigated within-urban and within-neighborhood pollution exposure differences (NO, NO_2_, BC, UFP) in the California Bay Area [22], finding that conclusions about exposure disparities differed depending on the chosen metric. First, in alignment with an “Equality” framing, they compared the central tendencies of the distribution of exposure for each population group. Second, they examined how the racial/ethnic composition of high-exposure and low-exposure groupings differed from the racial/ethnic composition of the total population. The examination of the population within the highest-exposure areas was complementary to the subgroup inequality metric of Stuart et al. but was more aligned with an “Exposure-driven” framing, placing the focus on which high-exposure areas and which pollutants were associated with the most disproportional exposure rather than which group experienced the greatest average burden, emphasizing distributional equity.

The fourth cross-sectional study by Thayer et al. (2022) investigated the association of the ultrafine particle count (UFP) with the local demographic composition and socioeconomic indicators in Boston, Massachusetts [23]. This approach differed considerably from Stuart et al. and Chambliss et al., as they applied single and multivariate regression modeling to examine the association of the UFP with variables including poverty, male unemployment, education, public assistance, home ownership, and female-headed households, as well as income and race. They conclude that high UFP levels were associated with demographics in complex ways; block groups with a predominately Asian ethnicity and lower income were significantly and positively related to a higher UFP concentration, while homeownership and a predominately Black population were both correlated with a lower UFP concentration. This provided a strong example of a “Socially weighted” framing, as the investigation focused on which demographic and SES aspects of a neighborhood were most associated with exposure risk.

### 3.2. Retrospective Longitudinal

The three longitudinal studies included here examined national trends in air pollution exposure over a decade or more, with the shared general aim of examining whether inequalities in air pollution exposure have lessened, and, to the extent that changes in air pollution over this time are attributable to previously enacted national policies, whether those policies advanced the EJ goals of both distributional and procedural equity.

Jbaily et al. (2022) focused on trends in PM_2.5_ concentrations across the US for the years 2000–2016 [4], finding inequalities along several dimensions: the population-weighted average concentration among Black Americans was higher than for other racial/ethnic groups; areas with higher-than-average white and indigenous populations were consistently lower in concentration; and lower income groups also experienced a higher average exposure. In examining areas above a threshold concentration of 8 mcg/m^3^, a greater share of the population in these areas was Black, and simultaneous patterns of change in concentrations and demographics showed that areas with increasing Black populations and increasing Hispanic/Latino populations also had increasing PM_2.5_ concentrations. The examination of trends over time and comparison of population-weighted concentrations supported “Improvement” and “Equality” conclusions, while the identification that the highest concentration areas continued to be disproportionately affecting specific racial/ethnic groups aligned with “Exposure-driven”. There was little examination of variability within racial/ethnic groups, although the focus on comparisons among Black and white Americans reflected external contextual factors aligned with a “Socially weighted” framework. The authors pointed to conclusions about procedural justice, a case of “policy not benefiting all areas equally” from the different rates of change in exposure in areas with different racial/ethnic compositions.

Colmer et al. (2020) also investigated trends in PM_2.5_ concentrations, covering the period of 1981–2016 [5]. Their analysis used both population-weighted averages and an approach focusing on the persistence of high concentrations in specific areas, identifying whether the ranking of census tracts by concentration changed from 1981 and 2016. While trends in population-weighted summary statistics showed an overall increase in exposure equality, the ranking analysis indicated ways that relative inequality persisted. Rankings at the extremes remained stable over time, showing that high-pollution areas remained the highest even as overall concentrations decreased. Racial/ethnic inequality was integrated into the ranking analysis by examining the statistical association of census tract demographics with changes in rank; among such associations, a higher share of white residents was associated with a percentile rank that decreased over time (relative conditions improved), while a higher share of Hispanic residents was associated with an increasing rank. The central framing of the analysis was “Equality”, but the identification of racial/ethnic groups that seemed to benefit less from improvements in air quality over time touched on issues of procedural equity aligned with a “Socially weighted” framework.

Liu et al. (2021) characterized changes in exposure and inequality for six pollutants (CO, NO_2_ O_3_, PM_2.5_, PM_10_, SO_2_) from 1990 to 2010, examining a wide range of inequality metrics [3]. Their core conclusions relied on population-weighted means; as air pollution levels declined, so did both absolute and relative inequality, but inequality was not eliminated across racial/ethnic groups, and across all years and pollutants, the highest average exposure occurred for racialized groups. These conclusions aligned with the “Equality” framing, further supported by both the Gini coefficient and Atkinson index to quantify overall population inequality. Supplemental analyses supported both “Exposure-driven” and “Socially weighted” conclusions. For the former, the finding that areas above the 90th percentile for multiple pollutants had disproportionately racialized populations suggested that intervention in these areas optimized both exposure and inequality reduction. For the latter, the stratification of racial/ethnic groupings by income—extending methods from Clark et al. (2014) [2]—sought to further identify specific SES features within racial/ethnic groups that may have contributed to a higher exposure burden.

### 3.3. Policy Projection

As one of the earliest studies to demonstrate methods of integrating equality concerns in air pollution policy impact assessments, Levy et al. (2007) modeled the effects of hypothetical emission controls on power plants to investigate whether some policies may disproportionately benefit a geographic subset of the US population, increasing inequality while decreasing overall exposure [24]. This work only evaluated total population inequality and did not examine differences by racial/ethnic or SES group, relying primarily on the Atkinson index. While the equity conclusions supported by their findings were limited, they did comment on an aspect of procedural equity tied to the “Exposure-driven” concept: “the focus of controls in geographic areas with elevated baselines means that benefits are spread less uniformly but serve to reduce existing disparities”.

Kelly et al. (2021) focused primarily on exposure assessment methods, including a projection of future of PM_2.5_ concentration based on policy commitments [25], with a secondary focus on whether conclusions about inequality are sensitive to the choice of exposure data. That analysis was limited to “Improvement” and “Equality” framings. They utilized the “risk gap” metric (difference between highest and lowest weighted average by race/ethnicity) to reduce the number of pairwise comparisons without introducing external judgment about which comparisons to prioritize, in contrast to the choice by Jbaily et al. (2022) to focus on the exposure gap between Black and white Americans [4]. They also quantified the degree of racial/ethnic inequality in each US state using the between-group Atkinson index (AI-BG). Although the authors did not explicitly address equity, they did examine whether projected exposure improvements varied by racial/ethnic groups; the non-Hispanic Black population was expected to experience the greatest reduction in exposure, although reductions were similar for white and Black populations on a percentage basis. The smallest air quality improvement was projected for the Hispanic group. This does raise questions of procedural equity in the “on the books” legislation underlying projected future concentrations.

The final policy projection analysis by Nguyen and Marshall (2018) used a selection of metrics to balance policies’ priorities of reducing total exposure, exposure inequality, and exposure injustice in a localized intervention in Los Angeles, California [26]. Nguyen and Marshall quantified inequality using the dissimilarity index (see Figure 2) and injustice using the difference in mean exposure for non-Hispanic white residents versus all others. Their analysis demonstrated a combination of both “Exposure-driven” and “Socially weighted” prioritization; a geographic area was identified for intervention in a top-down process based on exposure, but the choice of intervention was then informed by the social weighting of the distribution of benefits among residents.

## 4. Policy Implications and Recommendations for Implementation

The works reviewed here showcase the range of quantitative views of exposure inequality. Many studies feature multiple metrics, providing the intended audience—policy makers, environmental justice advocates, and researchers focused on environmental and health equity, among others—with a choice of results most aligned with their own equity goals [12]. However, it may be challenging for these stakeholders to draw specific conclusions about mitigation actions from a broad descriptive analysis in which the motivating ideals are not explicitly identified. Here, we contextualize the four framings as they extend to policies implemented in the US and provide recommendations for metric choices in future studies.

Measures of “Equal improvement” and “Equality” are most relevant to broad-scale, top-down reductions in pollution emissions, such as those typical of historical actions taken under the Clean Air Act, now over 50 years old. While these are not inherently measures of EJ, “Equality”-focused metrics evaluating whether policy outcomes align with basic distributional justice—that benefits accrue evenly across populations—may support as EJ a secondary goal.

“Exposure-driven” metrics may also be used with top-down targeting strategies, adding a population-focused dimension examining where emission reductions would most efficiently reduce exposure in high-concentration areas, with the primary goal of selecting among emission reduction options (e.g., electrification of freight carriers) to provide the greatest potential public health benefits [28]. To integrate EJ as a goal, decision-makers may also take into consideration which of the highest-pollution areas, such as nonattainment counties, are also disproportionately areas with people of color or low income. This primarily promotes distributional justice. A modification of this top-down structure could also advance procedural equity goals by applying such metrics to identify targets of outreach rather than predetermined interventions, providing local governing bodies and community-based EJ organizations with informational and financial resources to determine locally appropriate strategies for pollution reduction.

A combination of “Exposure-driven” and “Socially weighted” metrics align with interventions that focus resources on vulnerable populations, such as with the Justice40 Initiative, the federal government’s goal that 40 percent of certain investments go to communities that are overburdened with pollution, or California’s AB617 directive [29,30]. These policies are strengthened by an intersectional approach; rather than population-scale summary metrics which treat individuals as interchangeable, it is more useful to identify multiple characteristics associated with pollution exposure risk and use those to focus resources on the most overburdened groups. The groups themselves may be defined a priori based on historical discrimination or other known indicators of vulnerability/marginalization. Identifying these areas can facilitate procedural equity by directing policy makers and regulators to the places and people who will need to be brought into the policy-making process in order to satisfy the procedural equity goals of the EJ movement. “Socially weighted” metrics may also be used to inform urban planning decisions that would prevent the addition of polluting infrastructure in areas with vulnerable populations that are not yet considered highly exposed or that would allow the expansion of residential areas around existing polluters.

In future work identifying exposure inequalities with the intention of informing implementation of EJ interventions, we recommend providing complementary “Exposure-driven” and “Socially weighted” metrics, taking an intersectional view of areas and social groups that are both disproportionately impacted by pollution and are impacted by additional health risks resulting from structural racism. When considering exposure-priority methods to support procedural justice, it is useful to consider how the units of analysis may align with representative decision-making bodies. For example, a focus on geographic areas of both high exposure and historically marginalized communities could provide targets for outreach, or a focus on specific vulnerable communities could be used to facilitate the involvement of advocacy groups.

## 5. Conclusions

This work presents four conceptual framings of metrics of air pollution exposure inequity and highlights illustrative examples from the recent literature. As air pollution mitigation moves from documentation to taking action, metrics should match policy/intervention goals and be centered on EJ principles to maximize health equity. In this work, given the focus on empirical, quantitative research with US-centric policy considerations, distributional justice stands as the central focus, with some consideration of procedural justice but very little inclusion of recognitional justice [15]. The use of highly technical quantitative analyses such as those discussed here should be treated as one of many types of decision-making tools, and a respect for multiple ways of knowing should underly the interpretation and application of any of these metrics [31,32]. Additionally, while the focus of this paper is only the choice of metric in order to work toward greater EJ, there are a variety of other differences among these papers that may influence conclusions about improvements, equity, and effective prioritization, including the variety of air pollutants considered, the range of time over which exposures are tracked, and the geographic granularity of exposure data. While these technical limitations are not the focus of this commentary, they are also critical elements to consider.

## Figures and Tables

**Figure 1 ijerph-21-01706-f001:**
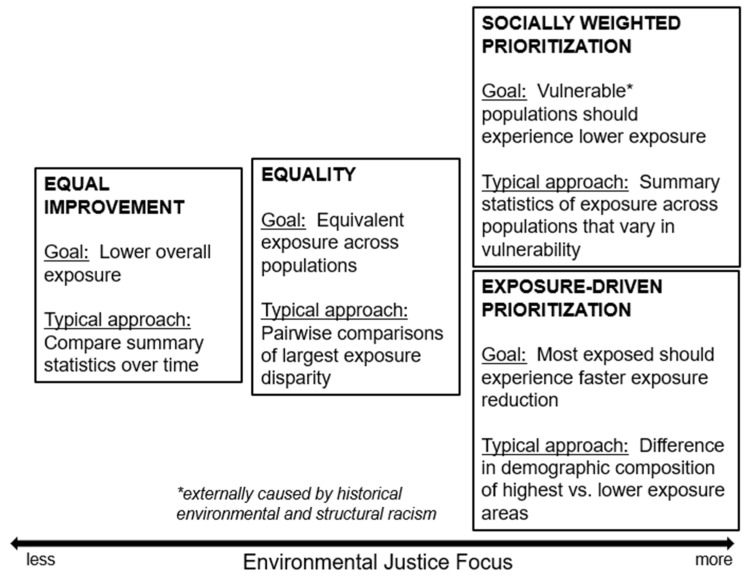
Summary of conceptual framings.

**Figure 2 ijerph-21-01706-f002:**
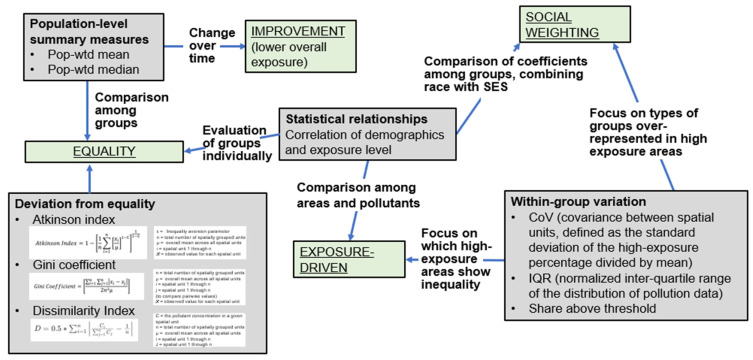
Metrics for evaluating exposure in an environmental justice framework. Green boxes show the four conceptual framings of environmental justice discussed here, gray boxes show metrics used to quantify exposure inequality, and labeled arrows indicate the application of those metrics to support each framework.

**Table 1 ijerph-21-01706-t001:** Summary of recently published manuscripts on air pollution inequities.

Author (Year)	Study Area/ Timeframe/ Pollutants	Comparisons by Race/Ethnicity and/or SES	Inequality Metric(s)	Conceptual Framing
Clark et al. (2014) [2]	US 2000, 2006 NO_2_	Both	Difference between low-income nonwhite people and high-income white people Atkinson index	Equality Socially weighted
Stuart et al. (2009) [21]	Hillsborough county, FL 1996–2005 CO, PM_10_, PM_2.5_, NO_2_, SO_2_	Both	Subgroup inequity index	Equality
Chambliss et al. (2021) [22]	San Francisco Bay Area 2010 UFP, NO, NO_2_	Race/ethnicity	Concentration distributions	Equality Exposure-driven
Thayer et al. (2022) [23]	Boston, MA 2013 UFP	Both	Pearson correlations	Socially weighted
Jbaily et al. (2022) [4]	US 2004–2016 PM_2.5_	Both	Weighted averages Atkinson index Gini coefficient Covariance	Equality Socially weighted
Colmer et al. (2020) [5]	US 1981–2016 PM_2.5_	Both	Rank–rank comparisons	Equality Socially weighted
Liu et al. (2021) [3]	US 1990, 2000, 2010 CO, PM_10_, PM_2.5_, NO_2_, SO_2_	Both	Absolute exposure gap Relative exposure gap Atkinson index Gini coefficient	Equality Exposure-driven Socially weighted
Levy et al.(2007) [24]	US 1999 SO2, NOx, PM_2.5_	Not included	Atkinson index Gini coefficient Mean log deviation Theil entropy index	Equality
Kelly et al. (2021) [25]	US 2011, 2028 SO2, NOx, VOC, PM_2.5_	Race/ethnicity	Atkinson index Exposure difference	Equality
Nguyen and Marshall (2018) [26]	Los Angeles, CA 2005 PM_2.5_	Race/ethnicity	Dissimilarity index Absolute exposure gap	Equality Exposure-driven Socially weighted

## Data Availability

No new data were created or analyzed in this study. Data sharing is not applicable to this article.

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
