# Peer review of "Alignment of Air Pollution Exposure Inequality Metrics with Environmental Justice and Equity Goals in the United States"

_ijerph, 2024, doi:10.3390/ijerph21121706_

Round 1

Reviewer 1 Report

Comments and Suggestions for Authors

In this study, the authors reviewed recent literature on air pollution exposure inequality with implications for EJ. They suggest a starting framework for distinguishing the values alignment among metrics and quantitative findings, and how those may extend to common policy approaches. They categorize the selection of papers chosen for this review by the type of study design: cross-sectional (n=4), retrospective longitudinal (n=3), and policy impact projection (n=3). Based on these articles, they describe inequality and inequity conclusions provided from several contrasting approaches.  Overall, the sample size is small (n=4+3+3), and a limited number may not provide comprehensive information. It is suggested to increase relevant literatures for discussion and analysis. In addition, it is suggested to verify and analyze the framework based on own research data.

Author Response

Comment 1: In this study, the authors reviewed recent literature on air pollution exposure inequality with implications for EJ. They suggest a starting framework for distinguishing the values alignment among metrics and quantitative findings, and how those may extend to common policy approaches. They categorize the selection of papers chosen for this review by the type of study design: cross-sectional (n=4), retrospective longitudinal (n=3), and policy impact projection (n=3). Based on these articles, they describe inequality and inequity conclusions provided from several contrasting approaches.  Overall, the sample size is small (n=4+3+3), and a limited number may not provide comprehensive information. It is suggested to increase relevant literatures for discussion and analysis. In addition, it is suggested to verify and analyze the framework based on own research data.

Response 1: Thank you for your thoughtful evaluation of this paper. As you point out, as a commentary/essay, this work does not provide a comprehensive review of the literature in the field of air pollution exposure disparity, but we have tried to include a selection of works that represent the range of approaches applied in such analyses. We have also edited the text to provide a more explicit mention of this limitation:

Lines 301-307: These works were chosen to illustrate how the range of metrics and approaches applied in current EJ analyses. This is not a comprehensive review, and thus we do not include all studies addressing inequality and environmental injustice in air pollution exposure, nor do we address all aspects of these works, but instead focus on elements of particular relevance to the framings described above.

Lines 488-490: In our review of recent literature on air pollution exposure inequality with implications for EJ, we have observed that studies The works reviewed here showcase the range of quantitative views of exposure inequality.

Reviewer 2 Report

Comments and Suggestions for Authors

This is an interesting commentary.  As such is is probably too long and could be significantly shortened.

As all of the data and context is US based, this MUST be reflected in a revised title.

There are no clear concise conclusions or recommendations presented in either the abstract or the paper, I strongly recommend that this be addressed and changed.

Towards the discussion section the referencing style changes somewhat, to using Author names and year (which is ok) but also include the reference numbers

Author Response

Comment 1: As such is is probably too long and could be significantly shortened.

Response 1: We appreciate your interest in the subject matter of our essay. We have edited throughout for both conciseness and clarity.

Comment 2: As all of the data and context is US based, this MUST be reflected in a revised title.

Response 2: Thank you for this observation. We have made the following revisions to the title:

Alignment of air pollution exposure inequality metrics with environmental justice and equity goals in the United States.

Comment 3: There are no clear concise conclusions or recommendations presented in either the abstract or the paper, I strongly recommend that this be addressed and changed.

Response 3: We have revised the abstract and the final section of the paper to clearly state the conclusions and to augment the discussion of policy implications and recommendations for implementation as follows:

Abstract, lines 29-35: We discuss these framings and describe inequality and inequity conclusions provided from several contrasting approaches drawn from recent work. We recommend that future work addressing environmental justice interventions include complementary “Exposure-driven” and “Socially-weighted” metrics, taking an intersectional view of areas and social groups that are both disproportionately impacted by pollution and are impacted by additional health risks resulting from structural racism, and consider implications for environmental justice beyond distributional equity.

Lines 552-595:

 In future work identifying exposure inequalities with the intention of informing implementation of EJ interventions, we recommend providing complementary “Exposure-driven” and “Socially-weighted” metrics, taking an intersectional view of areas and social groups that are both disproportionately impacted by pollution and are impacted by additional health risks resulting from structural racism. When considering exposure priority methods to support procedural justice, it is useful to consider how the units of analysis may align with representative decision-making bodies. For example, a focus on geographic areas of both high exposure and historically marginalized communities could provide targets for outreach, or a focus on specific vulnerable communities could be used to facilitate the involvement of advocacy groups.

Conclusions

This work presents four conceptual framings of metrics of air pollution exposure inequity and  highlights illustrative  examples from recent literature. As air pollution mitigation moves from documentation to taking action, metrics should match the policy/intervention goal and be centered on EJ principles to maximize health equity. In this work, given the focus on empirical quantitative research with US-centric policy considerations, distributional justice stands as the central focus, with some consideration of procedural justice, but very little inclusion of recognitional justice [15]. The use of highly technical quantitative analyses such as those discussed here should be treated as one of many types of decision-making tools, and a respect for multiple ways of knowing should underly the interpretation and application of any of these metrics [31, 32]. Additionally, while the focus of this paper is only the choice of metric in order to work toward greater EJ, there are a variety of other differences among these papers that may influence conclusions about improvements, equity, and effective prioritization, including the variety of air pollutants considered, the range of time over which exposures are tracked, and the geographic granularity of exposure data. While these technical limitations are not the focus of this commentary, they are also critical elements to consider.

Comment 4: Towards the discussion section the referencing style changes somewhat, to using Author names and year (which is ok) but also include the reference numbers.

Response 4: Reference numbers have been added, in particular for Jbaily et al, Liu et al., and Nguyen and Marshall.

Reviewer 3 Report

Comments and Suggestions for Authors

The present essay is a very interesting dissertation, the authors presented four conceptual framing approaches to metrics of air pollution exposure inequity and reviewed some recent literature to illustrate them. As air pollution mitigation moves from documentation to taking action, metrics should match the policy intervention goal and be centered on Environmental Justice principles to maximize health equity.

I just have one single observation:

The authors mentioned that "A basic goal in this approach is that the vulnerable population should not experience systematically higher exposure than the rest of the population. Rather, vulnerable populations should experience lower exposure because the health impact on them is greater due to the externally caused vulnerability". Nevertheless, How the vulnerable population is defined for them? Because some important features associated to air pollution health effects were not taken in account in this essay like sex, age, chronic diseases, pregnancy, et. 

Author Response

Comment 1: I just have one single observation: The authors mentioned that "A basic goal in this approach is that the vulnerable population should not experience systematically higher exposure than the rest of the population. Rather, vulnerable populations should experience lower exposure because the health impact on them is greater due to the externally caused vulnerability". Nevertheless, How the vulnerable population is defined for them? Because some important features associated to air pollution health effects were not taken in account in this essay like sex, age, chronic diseases, pregnancy, et. 

Response 1: Thank you very much for your interest in this work and this thoughtful observation. We have revised the text to specify our definition of vulnerable populations

Lines 253-268: The final framing approach (“Socially-weighted”) takes into account multiple characteristics impacted by structural racism that shape social determinants of health (e.g., considering low-income members of a racialized group), akin to an intersectional approach, when determining where changes in exposure improve equity the most. Building on the “Equality” concept that greater improvements are needed for higher-exposure groups, it further centers on EJ by using context beyond air pollution exposure to identify a population that is more vulnerable to the health effects of pollution exposure. A basic goal in this approach is that the vulnerable population should not experience systematically higher exposure than the rest of the population. Rather, vulnerable populations should experience lower exposure because the health impact on them is greater due to the externally caused vulnerability. Other individual characteristics affecting susceptibility to pollution-related health impacts, including sex, age, pregnancy, etc., may be considered when identifying vulnerable groups, but do not reflect the social prioritization of historically marginalized populations.

Reviewer 4 Report

Comments and Suggestions for Authors

The paper is well written and interesting to read, however I see the f issues that should be resolved before publishing this paper;

1.Line 142-154; Authors should provide more details of  "  low income and high income" or add the definition.

2. Authors shold provide initial guidance for more  applications of this methodology in implementation research.

Author Response

Comment 1.Line 142-154; Authors should provide more details of  "  low income and high income" or add the definition.

Response 1:

Instead, this framing presents equality as an equivalent reduction in exposure for minoritized populations and non-minoritized population, or for lower vs. higher income (or SES) groups, with “low income” defined by the ratio of household earnings to the federal poverty level.

We also clarify in Lines 324-326:

The studies also vary in the language and specific groupings used for race and ethnicity and “low income”, so some inconsistencies remain in the terminology used to describe race/ethnicity and income strata.

Comment 2. Authors should provide initial guidance for more  applications of this methodology in implementation research.

Response 2: We have revised both the abstract and final section of the paper to provide specific guidance:

Abstract, lines 29-35: We discuss these framings and describe inequality and inequity conclusions provided from several contrasting approaches drawn from recent work. We recommend that future work addressing environmental justice interventions include complementary “Exposure-driven” and “Socially-weighted” metrics, taking an intersectional view of areas and social groups that are both disproportionately impacted by pollution and are impacted by additional health risks resulting from structural racism, and consider implications for environmental justice beyond distributional equity.

Lines 552-561:

 In future work identifying exposure inequalities with the intention of informing implementation of EJ interventions, we recommend providing complementary “Exposure-driven” and “Socially-weighted” metrics, taking an intersectional view of areas and social groups that are both disproportionately impacted by pollution and are impacted by additional health risks resulting from structural racism. When considering exposure priority methods to support procedural justice, it is useful to consider how the units of analysis may align with representative decision-making bodies. For example, a focus on geographic areas of both high exposure and historically marginalized communities could provide targets for outreach, or a focus on specific vulnerable communities could be used to facilitate the involvement of advocacy groups.
